# Affect and exertion during incremental physical exercise: Examining changes using automated facial action analysis and experiential self-report

**Sinika Timme**, **Ralf Brand** *

Sport and Exercise Psychology, University of Potsdam, Potsdam, Germany

* ralf.brand@uni-potsdam.de

## Abstract

Recent research indicates that affective responses during exercise are an important determinant of future exercise and physical activity. Thus far these responses have been measured with standardized self-report scales, but this study used biometric software for automated facial action analysis to analyze the changes that occur during physical exercise. A sample of 132 young, healthy individuals performed an incremental test on a cycle ergometer. During that test the participants' faces were video-recorded and the changes were algorithmically analyzed at frame rate (30 fps). Perceived exertion and affective valence were measured every two minutes with established psychometric scales. Taking into account anticipated inter-individual variability, multilevel regression analysis was used to model how affective valence and ratings of perceived exertion (RPE) covaried with movement in 20 facial action areas. We found the expected quadratic decline in self-reported affective valence (more negative) as exercise intensity increased. Repeated measures correlation showed that the facial action *mouth open* was linked to changes in (highly intercorrelated) affective valence and RPE. Multilevel trend analyses were calculated to investigate whether facial actions were typically linked to either affective valence or RPE. These analyses showed that *mouth open* and *jaw drop* predicted RPE, whereas (additional) *nose wrinkle* was indicative for the decline in affective valence. Our results contribute to the view that negative affect, escalating with increasing exercise intensity, may be the body's essential warning signal that physiological overload is imminent. We conclude that automated facial action analysis provides new options for researchers investigating feelings during exercise. In addition, our findings offer physical educators and coaches a new way of monitoring the affective state of exercisers, without interrupting and asking them.

## 1. Introduction

Exercise plays a significant role in reducing the risk of developing diseases and in improving health and wellbeing [1], yet despite knowing that exercise is good for them most adults in Western countries are insufficiently active [2].

**Data Availability Statement:** The data underlying the results presented in the study are available from: https://osf.io/z8rv7/

**Funding:** The authors received no specific funding for this work.

**Competing interests:** The authors have declared that no competing interests exist.

Exercise psychologists have spent the last 50 years developing and testing theories about why some people are more successful than others in changing their behavior to promote their own health and exercise more regularly. After decades of focusing on social-cognitive factors and the role of deliberate reasoning in motivation (e.g. goal-setting and self-efficacy) researchers began to focus on the role of more automatic and affective processes in promoting change in health-related behaviors [3, 4, 5].

Affect has been defined as a pleasant or unpleasant non-reflective feeling that is always accessible and is an inherent aspect of moods and emotional episodes, but can be experienced independently of these states as well [6]. Affect can be described in the two orthogonal dimensions: 'affective valence' (how good or bad one feels) and 'arousal' (high vs. low) [7]. There is conclusive evidence that those who experience a more pleasant affective state during exercise are more likely to exercise again [8].

Dual-mode theory [9] explains how feelings during exercise are moderated by exercise intensity. According to the theory and supported by evidence [10], the affective response to *moderate* intensity exercise (below ventilatory threshold; VT) is mostly positive, but affective responses to *heavy* intensity exercise (approaching the VT) are more variable. Some individuals continue to report positive affect as exercise intensity increases, but others report more and more negative affect. When the intensity of exercise increases to the *severe* domain (when the respiratory compensation threshold, RCT, is exceeded), almost all individuals report a decline in pleasure [9, 10].

Ratings of affective valence above the VT are closely connected to the concept of perceived exertion. Borg [11, p. 8] defined perceived exertion as "... the feeling of how heavy and strenuous a physical task is". A recent article in *Experimental Biology* proposed that at high exercise intensities feelings of negative affect and perceived exertion may even convert into one, suggesting that the sensation of severe exertion enters consciousness via a decline in pleasure [12].

We believe that gaining a deeper understanding of the relationship between the affective response to exercise and perceived exertion is important not just from a research perspective, but also from a practical perspective. Practitioners (e.g. teachers and coaches) would greatly and immediately benefit from being able to assess an exerciser's perceived exertion and his or her momentary affective state to increase the odds of further effective and pleasurable physical exercise.

## 1.1 Measurement of exercise-induced feelings during exercise

Thus far exercise-induced feelings have been mostly measured with exercisers' self-reports [3]. The most commonly used psychometric measures of affective valence is the Feeling Scale (FS) [13], a single-item measure consisting of the question "How do you feel right now?" to which responses are given using an 11-point bipolar rating scale. Various studies have shown that displeasure increases with a quadratic trend under increasing exercise intensity, although with considerable inter-individual variability [10].

Perceived exertion, on the other hand, has often been measured with Borg's rating of perceived exertion (RPE) scale [11]. In this test participants are asked to indicate their actual state during exercise on a 15-point scale ranging from 6 *no exertion* to 20 *maximal exertion*. The scale is designed to reflect the heart rate of the individual before, during and after physical exercise. It would be assumed that an RPE of 13 corresponds approximately to a heart rate of 130 [14].

Focusing on two tasks simultaneously (exercising and rating one's own feelings at the same time) can bias the validity of the answer as well as the feeling states itself. It is known that the act of labeling affect can influence the individual's affective response [15]. Another limitation

is that affective valence changes during exercise [10] and repeatedly asking people how they feel inevitably carries the risk that it will interrupt their experience and introduce additional bias to their answers. Monitoring changes in biometric data avoids these interruptions and can thereby provide an alternative way to learn about the feelings that occur during exercise.

## 1.2 Facial action (facial expression) analysis

Spectators and commentators on sport readily infer how athletes might feel from their facial movements during exercise. Some of these "expressions" might reveal information about an athletes' inner state. However, it cannot universally be assumed that observed facial movements always reflect (i.e., are expressive of) an inner state [16]. Facial actions can also be related to perceptual, social, attentional, or cognitive processes [17, 18]. Therefore, we refer to facial expressions as facial actions in order to discourage the misunderstanding that subjective inner states are unambiguously expressed in the face.

The majority of studies conducted so far has quantified facial action by using either facial electromyographic activity (fEMG) or specific coding systems, of which the Facial Action Coding System (FACS) is probably the most widely known [19, 20].

fEMG involves measuring electrical potentials from facial muscles in order to infer muscular contractions. It requires the placement of electrodes on the face and thus can only measure the activity of a pre-selected set of facial muscles. Another limitation of using fEMG is that it is affected by crosstalk, meaning that surrounding muscles interfere with the signals from the muscles of interest, making fEMG signals noisy and ambiguous [21, 22]. A few fEMG studies have demonstrated that contraction of specific facial muscles (corrugator supercilii, zygomaticus and masseter muscle) is correlated with RPE during resistance training [21, 23] and bouts of cycling [20, 24].

Furthermore there are coding systems. Many of them are rooted in the FACS, which is an anatomy based, descriptive systems for manually coding all visually observable facial movements [19]. Trained coders view video-recordings of facial movements frame-by-frame in order to code facial movements into action units (AUs). FACS is time-consuming to learn and use (approximately 100 hours to learn FACS and one to two hours to analyze just one minute of video content) [20].

Recent progress has been made in building computer systems to identify facial actions and analyze them as a source of information about for example affective states [25]. Computer scientists have developed computer vision and machine learning models, which automatically decode the content of facial movements to facilitate faster, more replicable coding. The computer systems display high concurrent validity with manual coding [26].

We are aware of only one study so far that has used automated facial feature tracking to describe how facial activity changed with exercise intensity [27]. The authors analyzed video-recordings of overall head movement and 49 facial points with the IntraFace software to classify movement in the upper and lower face. The study showed that facial activity in all areas differed between intensity domains. The movement increased from lactate threshold until attainment of maximal aerobic power with greater movement in the upper face than in the lower face at all exercise intensities.

## 1.3 This study

The aim of this study was to examine changes in a variety of discrete facial actions during an incremental exercise test, and relate them to changes in self-reported RPE and affective valence, i.e. feelings that typically occur during exercise. To the best of our knowledge it is the first study to involve the use of automated facial action analysis as a method of investigating the covariation of these variables.

We have used an automated facial action coding system with the Affectiva Affdex algorithm at its core [28]. It includes the Viola Jones Cascaded Classifier algorithm [29] to detect faces in digital videos, and then digitally tags and tracks the configuration of 34 facial landmarks (e.g., nose tip, chin tip, eye corners). Data is fed into a classification algorithm which translates the relative positions and movements of the landmarks into 20 facial actions (e.g., *mouth open*). Classification by Affectiva Affdex relies on a normative data set based on manual initial codings of human FACS coders, and subsequent machine learning data enrichment with more than 6.5 million faces analyzed [30]. Facial actions as detected by Affectiva Affdex are similar [31] but not identical to the AUs from the FACS. Facial actions consist of single facial movements or combinations of several movements (e.g., facial action *mouth open*: lower lip drops downwards as indicated by AU 25 *lips part*; facial action *smile* as indicated by AU 6 *cheak raiser* together with AU 12 *lip corner puller*).

Connecting with dual mode theory [9] and research pointing out the importance of positive affect during exercise for further exercising [8], facial action metrics might provide useful biometric indicators for evaluating feeling states during exercise at different intensities. We took a descriptive approach to analyze which facial actions co-occur with affective valence and perceived exertion during exercise. This approach enables us to contribute conceptually to the examination of the relationship between the constructs of perceived exertion and affective valence (e.g. to determine if they are one or two distinct constructs and whether this depends on physical load) [12], whilst avoiding bias caused by repeatedly interrupting subjects' experience of exercise to obtain self-reports.

In order to account for expectable high inter- and intra-individual variability in both the affective response to exercise [3] and in facial actions [16], we used multilevel regression modeling to analyze our data; as far as we know, we are the first in this research area to use this method of data analysis.

## 2. Method and materials

The Research Ethics Committee of the University of Potsdam approved the study and all procedures complied with the Helsinki declaration. All participants gave their signed consent prior to partaking in the experiment. The individual in this manuscript has given written informed consent (as outlined in PLOS consent form) to publish these case details.

### 2.1 General setup

Study participants completed an exercise protocol involving exercising at increasing intensity on a cycle ergometer until they reached voluntary exhaustion. Whilst they were exercising their face was recorded continuously on video. Both affective valence and perceived exertion were measured repeatedly every two minutes. Changes in facial action were then evaluated with the help of software for automated facial action analysis and related to the self-report data. Advanced statistical methods were used for data analysis, accounting for the generally nested data structure (repeated measurements are nested within individuals).

### 2.2 Participants

We tested a group of 132 healthy individuals, aged between 18 and 36 years ($M_{age}$ = 21.58, $SD_{age}$ = 2.93; 53 women). All of them were enrolled in a bachelor's degree course in sport and exercise science. The group average of (self-reported) at least moderate physical activity was 337 minutes per week. Students with a beard or dependent on spectacles were not eligible to participate. Data from 19 participants were unusable due to recording malfunction ($n$ = 6), poor video quality ($n$ = 6; more than 10% missing values because the software did not detect

the face) or due to disturbing external circumstances ($n = 7$; people entering the room unexpectedly; loud music played in the nearby gym). This resulted in a final sample of 113 study participants.

### 2.3 Treatment and measures

**2.3.1 Exercise protocol.** The participants performed an incremental exercise test on an indoor bike ergometer. Required power output was increased by 25 watt increments every two minutes, starting from 25 watts until the participants indicated that they had reached voluntary exhaustion [32]. The protocol was stopped when the participant was unable to produce the required wattage any more. If a participant reached 300 watts, the final phase involved pedaling at this level for two minutes. Thus the maximum duration of the exercise was 26 minutes. All participants performed a five-minute cool-down consisting of easy cycling.

For a plausibility check whether self-declared physical exhaustion would be at least close to the participants' physiological state heart rate during exercise was monitored in about half of the participants ($n = 54$). A Shimmer3 ECG device with a sampling rate of 512 Hz was used for that. These participants started with a one-minute heart rate baseline measurement before the exercise.

**2.3.2 Affective valence and perceived exertion.** The FS (a single item scale: response options range from -5 *very bad* to +5 *very good*) [13] was used to measure affective valence, and participants rated their level of exertion using Borg's Rating of Perceived Exertion (RPE; a single-item scale; response options range from 6 *no exertion* to 20 *maximal exertion*) [11]. FS and RPE were assessed every two minutes during the exercise task, at the end of each watt level. For this purpose the two questionnaires (FS first and RPE second) were displayed on the monitor in front of the participants (see below) and they were asked to give their rating verbally to the experimenter.

**2.3.3 Automated facial expression analysis.** The participants' facial actions during the exercise task were analyzed using the software Affectiva Affdex [28] as implemented in the iMotions™ platform for biometric research (Version 7.2). Faces were continuously recorded with a Logitech HD Pro C920 webcam at a sampling rate of 30 fps during performance of the exercise task. The camera was mounted on top of the ergometer screen (0.4 m in front of the face with an angle of 20 degrees from below) and connected to the investigator's laptop.

Affectiva Affdex continuously analyzed the configuration of the 34 facial landmarks [31] during performance of the exercise task (Fig 1). It provided scores for 20 discrete facial actions (e.g., *nose wrinkle*, *lip press*) from all over the face (all facial actions detected by Affectiva Affdex are listed in Table 1, in the results section) [31]. The algorithm performs analysis and classification at frame rate. This means that at a time resolution of 30 picture frames per video

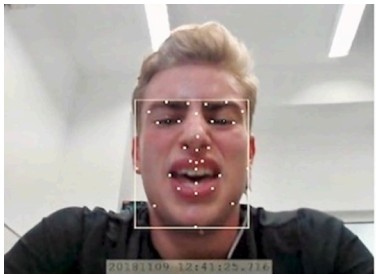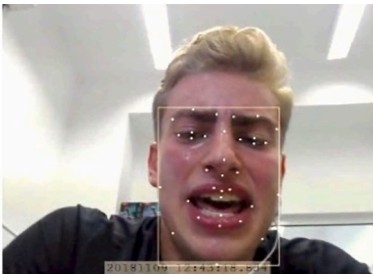

**Fig 1. Examples of facial actions during exercise.** Mouth open and nose wrinkle (left picture), jaw drop (right picture).The position of the 34 analyzed facial landmarks are marked with yellow dots.

**Table 1. Repeated measures correlation of all facial actions with FS and RPE.**

| Facial Action | FS | RPE |
|---|---|---|
| mouth open | -0.55* | 0.70* |
| jaw drop | -0.40* | 0.51* |
| nose wrinkle | -0.34* | 0.29* |
| lip pucker | -0.32* | 0.32* |
| upper lip raise | -0.31* | 0.27* |
| lid tighten | -0.30* | 0.26* |
| eye closure | -0.29* | 0.30* |
| smile | -0.26* | 0.25* |
| lip stretch | -0.21* | 0.18* |
| cheek raise | -0.19* | 0.21* |
| lip press | -0.19* | 0.15* |
| dimpler | -0.17* | 0.13* |
| brow furrow | -0.14* | 0.17* |
| eye widen | 0.14* | -0.27* |
| lip corner depressor | -0.13* | 0.13* |
| lip suck | -0.10* | 0.01 |
| inner brow raise | -0.10* | 0.07 |
| brow raise | -0.07 | 0.16* |
| chin raise | -0.07 | 0.03 |
| smirk | -0.07 | -0.06 |

FS, Feeling Scale; RPE, Rating of perceived exertion

*p < .001

second (30 fps), our analyses were based on 1.800 data points per facial action per 1 minute. Recent research has shown that Affectiva Affdex facial action scores are highly correlated with fEMG-derived scores, and that Affectiva Affdex outperforms fEMG in recognizing affectively neutral faces [33].

Each data point that Affectiva Affdex provides for a facial action is the probability of presence (0–100%) of that facial action. We aggregated these raw data, for each facial action separately, to facial actions scores (time percent scores) indicating how long during a watt level on the ergometer (i.e., within 2 minutes) a facial action was detected with the value 10 or higher. For example, a facial action score of 0 indicates that the facial action was not present during the watt level, whereas a score of 100 indicates that it was present all the time during that watt level.

Fig 1 illustrates examples of facial actions and the analyzed facial landmarks.

## 2.4 Procedure

After the participants arrived at the laboratory they were informed about the exercise task and told that their face would be filmed during the task. They were also given a detailed description of the two scales (FS and RPE), what they are supposed to measure and how they would be used in the study.

Participation was voluntary and all participants completed data protection forms and were checked for current health problems. Participants performed the exercise task on a stationary cycle ergometer in an evenly and clearly lit laboratory in single sessions. An external 22" monitor was positioned 1.5 m in front of the participant; this was used to display instructions during

the exercise session (instruction on watt level for 100 s always at the beginning of each watt level; the two scales, FS and RPE, always for 10 s at the end of each level). Throughout the trial, no verbal encouragement or performance feedback was provided and the researcher followed a standard script of verbal interaction. During the exercise session the researcher remained out of the participants' sight and noted the participant's verbal responses when FS and RPE responses were solicited. The periods during which participants were reporting their ratings were cut from the video for the facial action analysis.

## 2.5 Statistical approach, modeling and data analysis

Multilevel models were used to assess the anticipated increase in negative affect during exercise and to examine the relationships between facial action, affective valence and perceived exertion. We had multiple observations for each participant (20 facial action scores, FS, RPE), so that these repeated measurements (level 1) were nested within individuals (level 2). The main advantages of multilevel models are that they separate between-person variance from within-subject variance, so that estimates can be made at individual level as well as at sample level [34]. Because they use heterogeneous regression slopes (one regression model for each participant) multilevel statistics enable analysis of dependent data and a potentially unbalanced design (series of measurements with different lengths); two conditions that would violate test assumptions of traditional regression and variance analysis.

Our first model tested whether affective valence (FS) showed the expected quadratic trend [10] with increasing perceived exertion (RPE; time-varying predictor). In this model, RPE and derived polynomials were centered at zero and used as a continuous covariate for prediction of change in affective valence (FS).

To investigate which facial actions were associated with affective valence (FS) and with perceived exertion (RPE) we carried out separate analyses of the degree of covariation of FS and RPE with each facial action. First we looked at repeated measure correlations, which take the dependency of the data into account by analyzing common intra-individual associations whilst controlling for inter-individual variability [35]. Then we predicted affective valence (FS) from facial action whilst controlling for the influence of RPE, considering each facial action in a separate model. In parallel analyses we predicted RPE from facial action whilst controlling for the influence of FS. The significance of the fixed effects of facial actions were tested using chi-square tests for differences in -2 log likelihood values. A model with facial action as a predictor was compared with a reduced model without facial action. We compared all models in which FS or RPE was predicted by facial action, using the Akaike Information Criterion Corrected (AICC) and Weight of Evidence ($W$) [36]. Pseudo $R_x^2$ (within-subject level) was calculated to estimate the proportion of variance explained by the predictor [36].

Finally, to test whether FS and RPE made unique contributions in explaining variance in facial action, we calculated separate multilevel models in which specific facial actions were predicted by FS and RPE. This allowed us to partial out the separate amounts of explained variance of FS and RPE in the respective facial action.

We used the lme script from the nlme package (version 3.1–139) [37] to estimate fixed and random coefficients. This package is supplied in the R system for statistical computing (version 3.6.0) [38].

## 3 Results

### 3.1 Manipulation checks

As expected, participants reached different maximum watt levels in the exercise session and so the number of observations varied between participants. In summary, we recorded 1102 data

point observations for the 113 participants, derived from between 5 and 13 power levels per participant.

Mean maximum RPE in our sample was 19.29 ($SD$ = 1.01) and the mean heart rate in the final stage before exhaustion was 174.61 bpm ($SD$ = 16.08). This is similar to previously reported reached maximal heart rate in incremental cycling tasks (e.g. $HR_{max}$: 179.5 ± 20.2 bpm, in [39]). The correlation between heart rate and RPE was very high, $r$ = .82, $p$ < .001. We believe it is valid to assume that most of the participants were working at close to maximum capacity at the end of the incremental exercise session in our study.

### 3.2 Multilevel trend analysis of FS with RPE

An unconditional null model was estimated to calculate the intraclass correlation for affective valence (FS) ($\rho_I$ = .33), supporting the rationale of conducting multilevel analysis [34]. Next we introduced centered RPE (RPE_0) as a time-varying covariate to test the trend of FS with increasing RPE_0.

The model with a quadratic trend ($b_1$ = -0.01, $p$ = .65; $b_2$ = -0.02, $p$ < .001) provided a significantly better fit to the data compared to the linear model, $\chi^2$ (1) = 93.39, $p$ < .001. The inclusion of random slopes ($\chi^2$ (2) = 141.46, $p$ < .001) and random curvatures further improved the model fit significantly, $\chi^2$ (3) = 28.39, $p$ < .001. The full model, with RPE_0 and (RPE_0)$^2$ as fixed effects and random intercepts and slopes, explained 67.12% of the variance in FS.

Thus our results confirm previous results, indicating that FS showed the expected negative quadratic trend [11] with increasing intensity (RPE). Fig 2 illustrates the finding, which can be made particularly obvious by means of multilevel regression analysis: The high interindividual variability in the decrease of affective valence (more negative) under increasing perceived exhaustion is striking.

### 3.3 Repeated measures correlations

**3.3.1 Covariation of FS and RPE with facial action as intensity increases.** First correlations between each facial action and FS and RPE were calculated (**Table 1**). Repeated measures correlations revealed that *mouth open* ($r$ = -.55, $p$ < .001), *jaw drop* ($r$ = -.40, $p$ < .001) and *nose wrinkle* ($r$ = -.34, $p$ < .001) showed the highest correlations with affective valence (FS). *Mouth open* ($r$ = .70, $p$ < .001) and j*aw drop* ($r$ = .51, $p$ < .001) also showed the highest correlation with perceived exertion (RPE), followed by *lip pucker* ($r$ = .32, $p$ < .001). FS and RPE were highly correlated ($r$ = -.74, $p$ < .001). These results indicate that both FS and RPE were associated with *mouth open* and *jaw drop*.

### 3.4 Multilevel analyses

**3.4.1 Predicting FS from facial action whilst controlling for RPE.** To identify which facial action best explains variation in FS during an incremental exercise session we calculated separate multilevel models, one for each facial action (left column of Table 2). RPE was included in these models as a control variable with random intercepts and slopes. The model with *nose wrinkle* as the predictor showed the best fit (AICC = 2770.08, $W$ = 1). Parameter estimates ($b$ = -0.09, $p$ < .001) indicate a linear decrease in FS with increasing *nose wrinkle*. Adding *nose wrinkle* as a fixed effect significantly improved the model fit ($\chi^2$(1) = 12.37, $p$ < .001) compared to the reduced model (RPE predicting FS). Adding *nose wrinkle* to this model as a random effect further improved model fit significantly, $\chi^2$(3) = 32.89, $p$ < .001. *Nose wrinkle* explained 15.51% of the within-subject variation in FS. *Smile* showed the next best fit (AICC = 2780.83, W = 0), with parameter estimates ($b$ = -0.03, $p$ < .001) indicating a linear

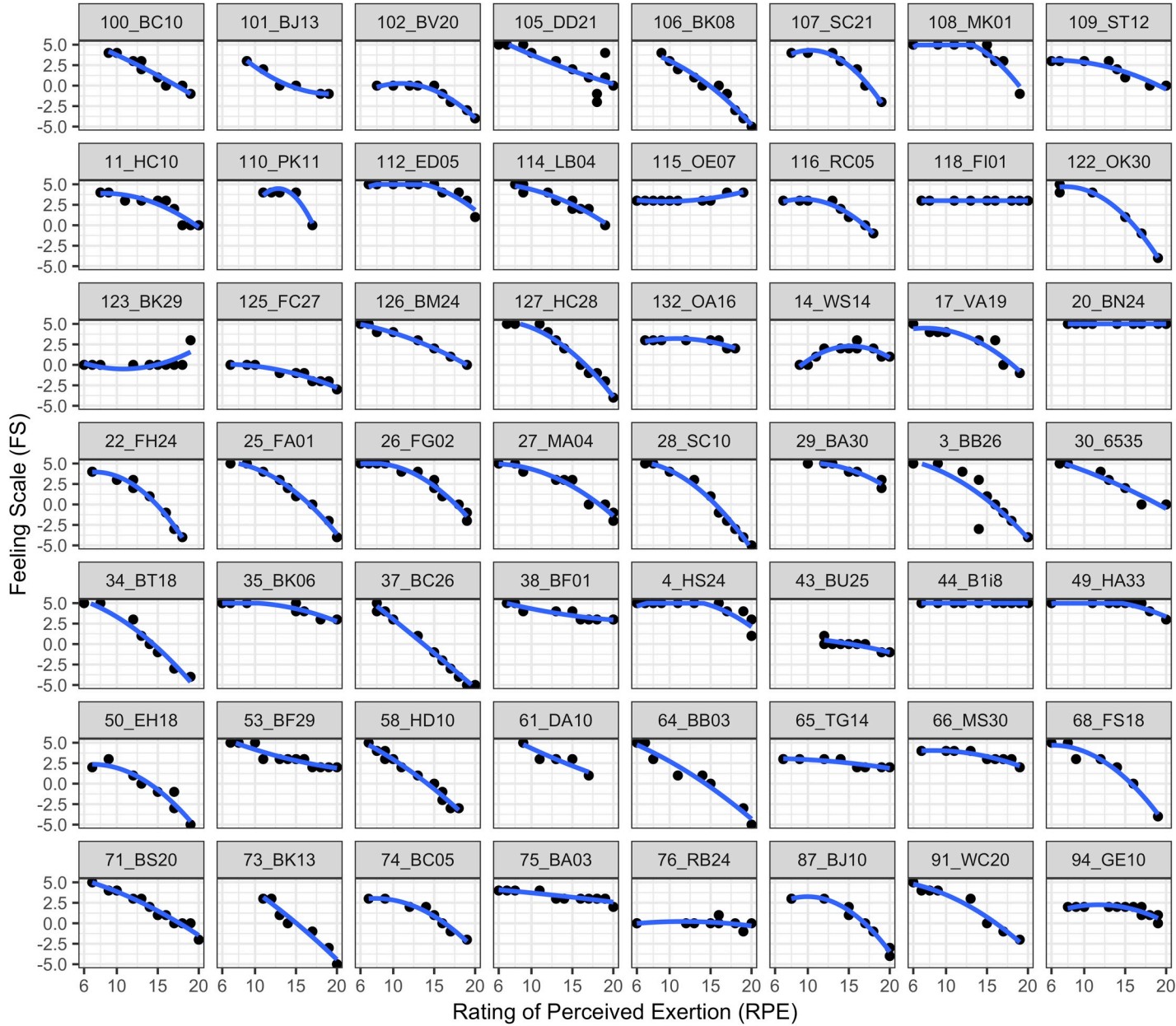

**Fig 2. Quadratic relationship between FS and RPE at individual level.** Data from a random selection of half of the participants ($n = 56$) are presented to illustrate the intra- and inter-individual variability in affective response to increasing exercise intensity. Intraclass correlation shows that 33% of the variance in affective valence (FS) is due to inter-individual variability.

decrease in FS as *smile* increased, explaining 3.47% of the within-subject variation in FS. All other facial actions showed an even worse model fit (left column of Table 2).

All in all, these results indicate that when controlling for the effects of RPE, *nose wrinkle* explains a significant proportion of the variation in affective valence and more than any other of the facial actions.

**3.4.2 Predicting RPE from facial action whilst controlling for FS.** To determine which facial action explains the most variation in RPE during the incremental exercise session we calculated a series of analyses in which RPE was predicted by all different facial actions in separate

**Table 2. Comparison of multilevel models in which one facial action predicts FS (left column) or RPE (right column).**

| model | FS | | | | RPE | | | |
|---|---|---|---|---|---|---|---|---|
| | K | AICC | Delta AICC | W | K | AICC | Delta AICC | W |
| reduced[a] | 7 | 2807.19 | 37.11 | 0 | 7 | 4192.92 | 57.86 | 0 |
| mouth open | 11 | 2796.18 | 26.10 | 0 | 11 | 4135.06 | 0 | 0 |
| jaw drop | 11 | 2810.51 | 40.43 | 0 | 11 | 4184.02 | 48.96 | 0 |
| nose wrinkle | 11 | 2770.08 | 0 | 1 | 11 | 4197.21 | 62.14 | 0 |
| lip pucker | 11 | 2805.21 | 35.13 | 0 | 11 | 4197.64 | 62.58 | 0 |
| upper lip raise | 8[b] | 2796.12 | 26.04 | 0 | 11 | 4199.05 | 63.99 | 0 |
| lid tighten | 11 | 2788.75 | 18.67 | 0 | 11 | 4200.74 | 65.68 | 0 |
| eye closure | 11 | 2810.74 | 40.66 | 0 | 11 | 4198.96 | 63.90 | 0 |
| smile | 11 | 2780.83 | 10.75 | 0 | 11 | 4200.38 | 65.32 | 0 |
| lip stretch | 11 | 2811.84 | 41.76 | 0 | 11 | 4197.89 | 62.82 | 0 |
| cheek raise | 11 | 2796.27 | 26.19 | 0 | 11 | 4199.80 | 64.74 | 0 |
| lip press | 11 | 2808.03 | 37.95 | 0 | 11 | 4198.29 | 63.23 | 0 |
| dimpler | 11 | 2813.96 | 43.88 | 0 | 11 | 4197.60 | 62.53 | 0 |
| brow furrow | 11 | 2803.04 | 32.96 | 0 | 11 | 4200.38 | 65.32 | 0 |
| eye widen | 11 | 2811.23 | 41.15 | 0 | 11 | 4188.29 | 53.23 | 0 |
| lip corner depressor | 8[b] | 2807.60 | 37.52 | 0 | 11 | 4200.32 | 65.25 | 0 |
| lip suck | 11 | 2814.52 | 44.44 | 0 | 11 | 4201.08 | 66.02 | 0 |
| inner brow raise | 11 | 2809.52 | 39.44 | 0 | 11 | 4200.38 | 65.32 | 0 |
| brow raise | 11 | 2813.85 | 43.77 | 0 | 11 | 4200.81 | 65.75 | 0 |
| chin raise | 11 | 2813.14 | 43.06 | 0 | 11 | 4200.92 | 65.86 | 0 |
| smirk | 11 | 2804.41 | 34.33 | 0 | 11 | 4197.28 | 62.22 | 0 |

FS, Feeling Scale; RPE, Rating of perceived exertion; K, number of parameters; AICC, Akaike information criterion corrected; W, weight of evidence. Models predicted FS (left column) resp. RPE (right column) with each facial action as a fixed and random factor while controlling for the influence of RPE resp. FS.

[a]The reduced model describes the respective outcome variable predicted by the respective covariate (left column: RPE predicting FS, right column: FS predicting RPE).

[b]The models with upper lip raise and lip corner depressor as a predictor of FS failed to converge. Therefore, a more parsimonious model without the facial action as a random factor was calculated, resulting in a smaller number of parameters (K).

multilevel models (right column of Table 2). FS was included in each model as a control variable with random intercepts and slopes.

Here *mouth open* showed the best model fit (AICC = 4135.06, $W = 1$), followed by *jaw drop* (AICC = 4184.02, $W = 0$). Parameter estimates for both *mouth open* ($b = 0.03$, $p < .001$) and *jaw drop* ($b = 0.02$, $p < .001$) indicate a linear increase in RPE with increasing facial action.

Adding *mouth open* as a fixed effect to the reduced model (FS predicting RPE) significantly improved model fit ($\chi^2(1) = 65.85$, $p < .001$) and this model explained 16.28% of within-subject variance in RPE. Adding *mouth open* as a random effect did not further improve model fit, $\chi^2(3) = 0.15$, $p = .99$.

Adding *jaw drop* as a fixed effect to the reduced model (FS predicting RPE) significantly improved the model fit ($\chi^2(1) = 16.84$, $p < .001$) and this model explained 5.37% of within-subject variance in RPE; adding *jaw drop* as a random effect did not further improve the model fit, $\chi^2(3) = 0.20$, $p = .98$.

All other facial actions showed a worse model fit (right column of Table 2), none explained more than 2.68% (*eye widen*) of the within-subject variation in FS.

Taken together these results indicate that *mouth open* and *jaw drop* explained significant variation in perceived exertion, and more than all other facial actions. Both facial actions involve movements in the mouth region; *jaw drop* is the bigger movement, as the whole jaw drops downwards, whereas *mouth open* only involves a drop of the lower lip [31].

**3.4.3 Predicting facial action from FS and RPE.** In order to separate the proportion of variance in the above identified facial actions (i.e. *mouth open* and *jaw drop*; *nose wrinkle*) explained by RPE and FS we calculated three separate multilevel models with each of these facial actions as the dependent variable and RPE and FS as time-varying predictors.

*Mouth open* was significantly predicted by both, RPE ($b$ = 2.53, $p$ = < .001) and FS ($b$ = -1.34, $p$ = .003). Introducing random slopes for RPE significantly improved model fit, $\chi^2$ (2) = 7.12, $p$ = .03. RPE accounted for 41.21% of the within-subject variance in *mouth open* and significantly improved the model compared to a reduced model without RPE as a predictor, $\chi^2$ (3) = 99.32, $p$ < .001. FS accounted for 11.42% of the within-subject variance in *mouth open* and significantly improved model fit compared with the reduced model without FS as a predictor, $\chi^2$ (1) = 10.50, $p$ = .001.

*Nose wrinkle* was significantly predicted by FS ($b$ = -0.32, $p$ = .003), but not by RPE ($b$ = 0.06, $p$ = .13). Introducing random slopes for FS and then RPE in separate steps significantly improved model fit; FS: $\chi^2$ (2) = 152.07, $p$ < .001, and RPE: $\chi^2$ (3) = 21.78, $p$ < .001. FS explained 21.10% of the within-subject variance in *nose wrinkle* and significantly improved model fit compared to the reduced model without FS as a predictor, $\chi^2$ (4) = 122.11, $p$ < .001.

*Jaw drop* was significantly predicted by RPE ($b$ = 1.06, $p$ < .001), but not by FS ($b$ = -0.49, $p$ = .12). Introducing random slopes for RPE significantly improved model fit, $\chi^2$ (2) = 14.54, $p$ < .001. RPE explained 35.83% of the within-subject variance in *jaw drop* and significantly improved model fit compared to a reduced model without RPE as a predictor, $\chi^2$ (3) = 59.02, $p$ < .001.

## 4 Discussion

The aim of this study was to examine whether and how single facial actions change with exercise intensity and how they were related to affective valence and perceived exertion. The study is innovative with regard to at least two aspects. First, we used automated facial action analysis technology to observe change in 20 discrete facial areas covering the whole face in a large sample of study participants. Second, the use of multilevel models allowed us to account for differences in change across individuals (nested data structure). We found that both affective valence and perceived exertion were significantly associated with *mouth open*. After controlling for the influence of RPE, *mouth open* was no longer significantly associated with affective valence, but the relationship between *mouth open* and RPE remained significant after controlling for the effect of affective valence. All in all, during exercise *nose wrinkle* was specifically characteristic of negative affect (i.e., less pleasurable feelings with increasing perceived exertion) and *jaw drop* of higher RPE. Fig 1 illustrates examples of these relevant facial actions.

### 4.1 Affective responses at different levels of perceived exertion

Several studies have investigated the change of affective responses during exercise with repeated measurement designs [10]. We think that this makes the separation of the intra- and inter-individual variability in data analysis inevitable. However, to the best of our knowledge there is currently no published study in which trajectories have been analyzed using the according multilevel regression approach. On the basis of dual-mode theory [9] and previous findings we hypothesized that there would be a negative quadratic trend [10, 40] of the affective response with increasing exercise intensity. Multilevel analysis confirmed this hypothesis and also demonstrated that there was high inter-individual variability in reported affective valence during exercise (Fig 2). This demonstrates the, in our view, necessity of using multilevel analysis when examining the decline in affect (more negative) during exercising with increasing intensity.

Previous studies were able to demonstrate the existence of inter-individual variability in affective valence by describing that e.g. 7% of participants reported an increase in affect ratings, 50% no change and 43% a decrease during exercise below the VT [41]. The statistical approach presented here extends this approach and allows to perform research that quantifies the influence of moderators of the exercise intensity-affect relationship to explain inter-individual differences in affective responses to exercise at given intensity level.

## 4.2 Affective responses and facial action

In our study affective valence was most highly correlated with the facial action *mouth open* when using simple repeated measures correlations (Table 1). However, affective valence was highly correlated with RPE, which was in turn highly correlated with *mouth open*. In order to determine what facial actions account for components of variance in specific constructs it is necessary to take into account the multicollinearity of the constructs. We did this by controlling statistically for variance in one construct (e.g. RPE) when analyzing the effect of the other (e.g. affective valence). When the influence of perceived exertion was taken into account, affective valence was most strongly associated with the facial action *nose wrinkle* (Table **2**). This is consistent with previous research showing that nose wrinkling may indicate negative affect. For example, newborns [42] and students [43] respond to aversive stimuli (e.g., a sour liquid [42] or offensive smells [43]) by wrinkling their nose. Perhaps pain is the context most relatable to high-intensity exercise. Studies of pain have identified nose wrinkling as an indicator of the affective dimension of pain [44], which is highly correlated with, but independent from, the sensory dimension [45].

*Nose wrinkle* has also been specifically associated with the emotion disgust [15]. However, the same facial action has been observed in various other situations (e.g. while learning) [46] and emotional states (e.g., anger) [47] and is not always observed concomitantly with reports of disgust [48]. *Nose wrinkle* may be indicative of negative affect more generally, rather than of a specific emotional state therefore.

*Nose wrinkle* explained more variance in affective valence than any other facial action, but given that this is the first study to have examined changes in facial action and affect during the course of an incremental exercise test and was performed with a sample of healthy adults, we suggest limiting the conclusion to the following: *nose wrinkle* is a facial action indicating negative affect in healthy adults during incremental exercise. To draw more general conclusions, for example, that *nose wrinkle* is the characteristic expression of negative affect during exercise, further research is needed. It should be demonstrated, for example, that this facial action reliably co-occurs with negative affect and that this co-occurrence prevails across several exercise modalities (e.g., running, resistance training).

## 4.3 Perceived exertion and facial action

The facial actions that were most highly correlated with perceived exertion, when controlling for the effect of affective valence were *mouth open* and *jaw drop* (Table **2**). On one hand, this is in line with research showing that activity in the jaw region is correlated with RPE [24]. At first sight, this may not go well with the findings from the fEMG study [22] that suggested that perceived exertion during physical tasks is mainly linked with corrugator muscle activity. It is important to note, however, that fEMG only measures activity in the muscles to which electrodes were attached (apart from noisy crosstalk), and that it cannot capture the dynamics of the whole face [49].

On the other hand, it is worth pointing out that we observed a correlation between RPE and *brow furrow* (which partly reflects corrugator activity). This correlation was smaller than

the two correlations between RPE and *jaw drop* and *mouth open* however (Table 1). First and foremost, it must be noted that as physical exertion increases, the exerciser is likely to breath heavier. The change from nose to mouth breathing is certainly to be interpreted against the background that more air can flow faster through the mouth. The observed change in facial action (i.e. increased *mouth open* and *jaw drop*) therefore most likely correlated with the physiological need for optimized gas exchange in the working organism. It is therefore particularly important to exploit the advantages of automated facial action analyses of the whole face and discrete facial actions to investigate the covariation of the various facial actions more closely.

## 4.4 Affective responses, perceived exertion and facial action

Both affective valence and perceived exertion were significantly associated with the facial action *mouth open* (Table **2**). While *nose wrinkle* was specific in explaining significant amounts of variance in affective valence and *jaw drop* in perceived exertion, *mouth open* explained significant amounts of variance in both affective valence and physical exertion (the facial action *mouth open* is described as "lower lip dropped downwards" in the Affectiva developer portal; *jaw drop* is "the jaw pulled downwards" with an even wider and further opening of the mouth [31]). This pattern of results might be interesting for the conceptual differentiation of affective valence and perceived physical exertion.

The two concepts, affective valence and physical exertion, are certainly closely linked [12]. This is reflected in our finding that the two are significantly correlated with the same facial action–*mouth open*. However, when the relationship of affective valence with the facial actions was controlled for the influence of RPE, *mouth open* explained only 1.19% of the within-subject variance in affective valence; *nose wrinkle* explained 15.51% on the other hand. These results suggest that mouth opening can be seen as a sign for the physical exertion portion in the experienced affect, whereas *nose wrinkle* indicates negative affect specifically.

*Jaw drop* (as the more extreme mouth opening), on the other hand, appeared not to be related to affective valence. *Jaw drop* could thus be assumed to be the more specific sign for (excessive) perceived exertion. Both the metabolic thresholds, VT and RCP, are related to perceived exertion. They are objective, individualized metabolic indicators of intensity, and are already associated with psychological transitions in dual mode theory [9]. Linking them to transitions in facial actions could be a future prospect and be something like this: While exercising at the VT might mark the transition between nose to (predominantly) mouth breathing and thus also the transition to more *mouth open*, exercising above the VT might mark a transition to more *jaw drop*. This kind of intensified breathing might covary with escalating negative affective valence–that is the evolutionary built-in warning signal that homeostatic perturbation is precarious and behavioral adaptation (reduction of physical strain) is necessary [12]. We have not analyzed the dynamics of the different facial actions in our study under this aspect, as this would not have been appropriate because we did not measure physiological markers for exercise intensity. But we suggest that future research should focus on exactly that.

## 4.5 Context- and individual-specific facial actions

This study can also be seen as a contribution to the current debate on what the face reveals about underlying affective states and whether universal, prototypical emotional facial expressions exist [16]. Our results support the notion that specific facial actions must be associated with affective states in a context- or individual-dependent manner in the first place. For example, *smile* (AU 6 + AU 12) is typically associated with the emotion "happiness" [50] and with positive affective valence [51]. This does not match our finding, and that of another study in the context of exercise [21], that *smile* can also be correlated with negative affective valence.

The use of biometric indices of facial action to measure psychological states requires that one takes into consideration that facial action is subject to high intra- and inter-individual variability [16]. Using multilevel analyses allowed us to take this into account. Due to the fact that some people show little or no movement in their faces, aggregational grand mean analyses such as a repeated measures ANOVA (which does not first model individual change) would be biased by this variation. Such analyses treat individual deviation from the grand mean as residual error, leading to the loss of important information about inter-individual differences. By taking individual trajectories into account, multilevel analyses allowed us to separate within-subject variance from between-subject variance and hence to adjust for obvious individual differences in facial action.

## 4.6 Limitations and recommendations for further research

Among the limitations of our study are the following: Basically we argued that automated facial action analysis could be an alternative for a more unobtrusive measurement of feelings during exercise. It is important not to lose sight of the fact, however, that simply knowing that you are being filmed can of course also change your behavior [52]. Another point is that although this study primarily focused on the correlations between facial actions and ratings of affective valence and perceived exertion, it would be advantageous to determine exercise intensity physiologically at the level of the individual participants in future studies (e.g., by the use of respiratory gas analysis in a pretest). This would have given us more confidence as to whether the majority of our participants have actually reached a state close to physical exhaustion at the end of the exercise protocol. Considering the participants' average RPE in their maximum watt levels and the comparison of the achieved heart rates with other studies on bicycle ergometers we think this is likely, but we cannot be sure of course. We further suggest that future studies should use more heterogeneous participant samples and a greater variety of sports and exercises to assure higher generalizability of the findings. Different modalities and different exercise intensities might produce specific facial actions. More heterogeneous samples are likely to produce more variance in affective responses, which may lead to further insight into the variation in facial reactions to exercise.

## 5 Conclusion

We conclude that both affective valence and perceived exertion can be captured using automated facial action analysis. Escalating negative affect during physical exercise may be characterized by nose wrinkling, representing the 'face of affect' in this context. The 'face of exertion', on the other hand, may be characterized by jaw dropping.

From a practical perspective, these results suggest that observing the face of an exerciser can give instructors important insights into the exerciser's momentary feelings. Facial actions can tell a lot about how the individual feels during exercise, and instructors could use individual facial cues to monitor instructed exercise intensity; to enhance exercisers' affective experience during exercise, which, at least for those who are not keen on exercise, is an important variable for maintaining the disliked behavior.

## Author Contributions

**Conceptualization:** Sinika Timme, Ralf Brand.

**Investigation:** Sinika Timme, Ralf Brand.

**Methodology:** Sinika Timme, Ralf Brand.

**Writing – original draft:** Sinika Timme, Ralf Brand.

**Writing – review & editing:** Sinika Timme, Ralf Brand.

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
