## [Decision Letter · Decision Letter 0]

22 Nov 2019

PONE-D-19-26758

Affect and exertion during aerobic exercise: Examining changes using automated facial expression analysis and continuous experiential self-report.

PLOS ONE

Dear Mrs. Timme,

Thank you for submitting your manuscript to PLOS ONE. After careful consideration, we feel that it has merit but does not fully meet PLOS ONE’s publication criteria as it currently stands. Therefore, we invite you to submit a revised version of the manuscript that addresses the points raised during the review process.

We would appreciate receiving your revised manuscript by Jan 06 2020 11:59PM. To enhance the reproducibility of your results, we recommend that if applicable you deposit your laboratory protocols in protocols.io, where a protocol can be assigned its own identifier (DOI) such that it can be cited independently in the future. For instructions see: http://journals.plos.org/plosone/s/submission-guidelines#loc-laboratory-protocols

We look forward to receiving your revised manuscript.

Kind regards,

Dominic Micklewright, PhD CPsychol PFHEA FBASES FACSM

Academic Editor

PLOS ONE

Reviewers' comments:

Reviewer's Responses to Questions

**Comments to the Author**

1. Is the manuscript technically sound, and do the data support the conclusions?

Reviewer #1: Yes

Reviewer #2: Yes

2. Has the statistical analysis been performed appropriately and rigorously? 

Reviewer #1: Yes

Reviewer #2: Yes

3. Have the authors made all data underlying the findings in their manuscript fully available?

Reviewer #1: No

Reviewer #2: Yes

4. Is the manuscript presented in an intelligible fashion and written in standard English?

Reviewer #1: Yes

Reviewer #2: Yes

5. Review Comments to the Author

Reviewer #1: Thank you for inviting me to review ‘Affect and exertion during aerobic exercise: Examining changes using automated facial expression analysis and continuous experiential self-report.’ The primary aim of this study was to examine continuous changes in single facial actions of the whole face at various exercise intensities. The authors concluded that affective valence and perceived exertion can be captured using automated facial action analysis.

This study uses novel methods and technology to assess changes in facial actions during exercise. I commend the authors on doing this innovative project and obtaining substantial participant numbers. Overall, I found the paper quite hard to read due to superlative words and lack of information provided about the methods. At present, the work is not accessible to non-specialists because of this, and improvements need to be made to make it more comprehensible. I think this will also make the implications of this project more obvious to the reader.

I have provided specific line-by-line comments, as well as two general comments below:

- Please provide the raw data as supplementary material (not video but the numbers produced from the video analysis (i.e. percentage changes in different facial actions)).

- Please provide more detail about the methods used. It is a good opportunity to provide a methods section that others can use to do further research, and in a journal such as PLOS ONE I would expect some more detail on this. At present your methods could not be replicated by the reader in future research.

Abstract

- Page 2, Line 2-6 – This sentence is quite hard to read and long. Clarity of your point may be made better by reducing words or splitting into two sentences.

- Page 2, Line 21 – Not sure about using (e.g.) in brackets as part of a flowing sentence. Maybe find a better way to write this?

Introduction

- Page 4, Line 2-6 – Why have you chosen ventilatory threshold and respiratory compensation threshold for these? Also, both need a reference.

- Page 5, Line 9 – I am not sure I agree with the comment that biometric data is more unobtrusive. I think videoing someone’s face and facial analysis in general is just as obtrusive; it is just different than techniques researchers have traditionally used before. I think it is more powerful, but not more unobtrusive. Would like to see this statement rectified.

- Page 6, Line 7 – FEMG should be fEMG

- Page 6, Line 11 – I don’t think you need to add an abbreviation for facial feature tracking as FFT. It wasn’t done in the original paper, isn’t commonly abbreviated and you don’t talk about it much after this.

- Page 7, Line 18 – That first statement is quite strong, and I believe there are more than two processes that have been used. Maybe tone it down a bit and just say, “the most commonly used are…”

- Throughout introduction (and the whole paper), you use facial action, facial expression and facial configurations interchangeably. I would just choose one, in your case it seems to be facial actions (especially considering you define why you are using this, which is good), and be consistent throughout.

- Generally, I am not sure about the structure of the introduction. The reader must read through nearly eight pages of background information before you get to the project aim. I think that some of this background content is of minor consequence to the rest of the paper, quite repetitive and could be made significantly briefer. You want the reader to get straight into the sections that are really related to the content of your paper, like parts of 1.3 and all of 1.4. For example, the following sections don’t really relate strongly to the paper and could be removed or shortened significantly:

o Page 3, Section 1 – except for the definition of affect on lines 8-12

o Page 4, Section 1.1

o Page 6, Paragraph 4

o Page 7, Paragraph 1 and 2

Method and materials

- Page 10, Line 1 – the use of the term modern is unnecessary. It is just software.

- Page 10, Line 4 – what do you define as fit? Do you have any information on training history or sports played? Did you perform any baseline cycling assessments?

- Page 10, Section 2.2 – Can you report participant height and weight?

- Page 10, Line 7 – Why was the video quality poor?

- Page 10, Line 8 – What happened in the environment to cause this?

- Page 10, Line 10 – “after one minute of baseline measurement” Baseline measurement of what?

- Page 10, Line 10-12 - A reference for the power output selection would be good.

- Page 10, Line 13-15 – “If a participant reached 300 watts the final phase involved pedaling at this level for two minutes, thus the maximum duration of the exercise was 26 minutes.” Why did you do this? How many participants completed the final stage without reaching volitional exhaustion?

- Page 10, Line 17-19 – Why did only half of participants have heart rate monitors?

- Page 11, Section 2.3.3 – I think this would be a really good section for a diagram showing the points on the face or a technical schematic of how the software works.

- Page 11, Section 2.3.3 – I think it would be good to have at least two cameras to pick-up different face angles. Is there a reason why you didn’t implement this? Do you think this affected the data you obtained during higher intensities when the head can drop? I think it is also important to know the distance the camera was to the face, an idea of head to camera angle, if it was adjusted for different participants height on the bike etc.

- Page 12, Section 2.3.3 – I think you need to describe more about the time each video frame was analysed for (i.e. milliseconds, seconds, batched into bigger 10 second blocks). At present it is not clear exactly how this was done.

- Page 12, Section 2.3.3 – What is threshold analysis?

- Page 12, Section 2.4 – So the participants knew the purpose of the video camera? I have a bit of an issue with this as I think this substantially changes how a person reacts to the video camera. Please address this as a limitation in your discussion. Might also be worth adding a reference for this manipulation of facial action when the participant knows their expression is being monitored. For example, this reference: Philippen P, Bakker F, Oudejans R, Canal-Bruland R. The Effects of Smiling and Frowning on Perceived Affect and Exertion While Physically Active. J Sport Behav. 2012;35(3):337-352.

Results

- Page 14, Line 13 – You are introducing some commentary and interpretation into your results, which I would suggest removing or putting in your discussion

- Page 14, Lines 17-19 - Based upon the fact that HR was only recorded for half participants and they only reached a mean HR of 174.61 and that RPE was 19 with an SD over 1 which suggests a large number of participants reported RPE well below maximum, I really don’t think you can say that for this age group they reached maximum capacity. Please alter statement.

- The image quality and clarity of Figure 1 is poor and does not convey the point that I can see you want to get across very well. Maybe think of an alternative way to represent this data?

Discussion and conclusion

- Page 23, Line 11 – you refer to the “aerobic exercise” here. This I the first time you refer to it as this, which is a bit odd. Maybe better to keep it consistent to what you said in the introduction, such as an incremental test.

- Page 23, Line 13 –you use e.g. mid-sentence again. Will flow better if written as for example or something similar in the sentence structure.

- Page 26, Section 4.6 – Generally, you don’t report any of the limitations that this study appears to have in your limitations section. Some of the main ones I have noted above are the fact participants knew the purpose of the video cameras so could change expressions, you didn’t obtain HR for all participants so cannot determine if they did actually reach the intensities you aimed for, and the final exercise intensity was absolute and fixed to 300W. Please rectify.

- Page 26, Lines 7-9 – This is a strong statement. I would just tone it down a bit…

- Page 26, Lines 10-11 – I like this section and agree with what you are saying, but I don’t feel the methods section you have provided in this paper would allow this research to be expanded upon by others. I think an improved methods section could be the strength of this paper

- General comment throughout discussion and conclusion– You discuss that mouth open and draw drop are the “face of exertion”. I don’t disagree with this statement per say, and yes, it is shown in your results. However, I feel you need to more strongly note that as RPE increases, someone is likely to be breathing heavier, and therefore the jaw drops and mouth opens. At present you don’t really discuss this in any detail, which I feel is an oversight in the interpretation of your findings.

Reviewer #2: Review of manuscript PONE-D-19_26758 submitted to PLOS ONE

S. Timme, R. Brand. Affect and exertion during aerobic exercise: Examining changes using automated facial expression analysis and continuous experiential self-report.

In this study, facial expressions during aerobic exercise were recorded at fixed time intervals using an algorithm to detect specific facial actions and identify them as actions units defined within the Facial Action Coding System (FACS). These expressions were related to subjective ratings of positive or negative affective state as well as ratings of perceived exertion. Covariations between facial expressions and subjective ratings were analyzed using multilevel regression or trend analysis, allowing investigation of these relationships at the level of individual participants.

The rationale behind this study is clear, the statistical analyses are sophisticated, and the manuscript is well written. I agree with the authors that a better insight in the exerciser's affective state and subjective feelings of exertion may contribute to stimulating people to engage in further pleasurable physical exercise. Although the results of this study seem clear, I have a few comments, questions, or (generally minor) concerns.

1. On pp. 6-7, and in the remainder of this manuscript, emphasis is laid on facial actions as indices of affective states or specific emotions. However, facial actions may also be related to perceptual, motivational, attentional, or cognitive processes (see, for example, studies mentioned in reference 58 of the current manuscript; see also Overbeek et al., 2014, Stekelenburg & van Boxtel, 2001). Although in the current study a lot of different facial actions were measured it seems as if the authors a priori consider these actions as indices of emotional processes whereas within FACS individual actions units strictly do not refer to specific emotions.

2. Although the current software used for analyzing facial actions indeed detected elementary facial actions as indicated in Table 1, it is remarkable that one of these actions ("smile") does not represent an elementary action but a combination of actions (AU6, check raiser; AU12, lip corner puller). This combination is generally interpreted as signifying a smile. I find this confusing since emotions are strictly not measured in the current study. On p. 25, it is said that in this study smile was associated with a negative affective valence. In line 12 on this page, it is erroneously suggested that the detection of AU6 is synonymous with the occurrence of a smile.

3. Figure 2 illustrates relevant facial actions which were observed during the current physical exertion task. I find these examples somewhat confusing since for the reader it may be difficult to associate them with an aerobic exercise task. But particularly the illustration of jaw drop is confusing since this facial expression also depicts AU's 6 and 12 which are generally considered to represent happiness, suggesting that this person is overtly laughing. I have shown this picture to several colleagues asking them to indicate what they saw. They reported to see an overtly laughing person.

4. On p. 23, it is defended that nose wrinkle need not be specifically related to disgust and that it may also be indicative of other emotions. However, in this respect studies are mentioned which have been performed in infants. I am afraid that facial expressions of infants cannot directly be compared with those of adults.

5. Later on this page, it is concluded that mouth open and jaw drop are highly correlated with perceived exertion but that this does not agree with results from an EMG study which would suggest that perceived exertion during physical tasks is mainly linked with corrugator activity. This brings me to the general question whether discrepancies between different studies may (at least partially) be related to studying either aerobic or anaerobic exercise. This distinction is not really discussed in this manuscript. When suggesting on p. 26, third paragraph, that future studies should include a wider ranger of sports to assure a higher generalizability of the current results, I wonder whether types of anaerobic exercise shouldn't also be included.

Minor points

- P. 7, line 3: "action" > "actions"

- P. 9, line 10: "action" > "actions"

- P. 11, line 11: "Logitech HD Pro C920" > "Logitech HD Pro C920 webcam"

- P. 19, footnote to Table 2: "in less number of parameters" > "in a smaller number of parameters"

References

Overbeek, T.J.M, van Boxtel, A., & Westerink, J.H.D.M. (2014). Respiratory sinus arrhythmia responses to cognitive tasks: Effects of task factors and RSA indices. Biological Psychology, 99, 1-14.

Stekelenburg, J.J., & van Boxtel, A. (2001). Inhibition of pericranial muscle activity, respiration, and heart rate enhances auditory sensitivity. Psychophysiology, 38, 629-641.

6. PLOS authors have the option to publish the peer review history of their article (what does this mean?). If published, this will include your full peer review and any attached files.

Reviewer #1: No

Reviewer #2: No

---

## [Author Response · Author response to Decision Letter 0]

8 Jan 2020

Reviewer 1

Reviewer #1: Thank you for inviting me to review ‘Affect and exertion during aerobic exercise: Examining changes using automated facial expression analysis and continuous experiential self-report.’ The primary aim of this study was to examine continuous changes in single facial actions of the whole face at various exercise intensities. The authors concluded that affective valence and perceived exertion can be captured using automated facial action analysis.

This study uses novel methods and technology to assess changes in facial actions during exercise. I commend the authors on doing this innovative project and obtaining substantial participant numbers. Overall, I found the paper quite hard to read due to superlative words and lack of information provided about the methods. At present, the work is not accessible to non-specialists because of this, and improvements need to be made to make it more comprehensible. I think this will also make the implications of this project more obvious to the reader.

Thank you very much for the overall positive evaluation and especially for your specific advice. We have tried to consider and revise all passages of the manuscript accordingly.

I have provided specific line-by-line comments, as well as two general comments below:

- Please provide the raw data as supplementary material (not video but the numbers produced from the video analysis (i.e. percentage changes in different facial actions)).

We made our data available on OSF (https://osf.io/z8rv7/) as requested by PlosOne. 

- Please provide more detail about the methods used. It is a good opportunity to provide a methods section that others can use to do further research, and in a journal such as PLOS ONE I would expect some more detail on this. At present your methods could not be replicated by the reader in future research.

Many thanks for this advice. We tried to thoroughly revise the text to this effect (and present all changes made point by point below).

Abstract

Page 2, Line 2-6 – This sentence is quite hard to read and long. Clarity of your point may be made better by reducing words or splitting into two sentences.

We split the sentence into two.

 Page 2, Line 21 – Not sure about using (e.g.) in brackets as part of a flowing sentence. Maybe find a better way to write this?

We corrected this.

Introduction

- Page 4, Line 2-6 – Why have you chosen ventilatory threshold and respiratory compensation threshold for these? Also, both need a reference.

We chose the ventilatory and respiratory compensation threshold because the two are explicit aspects of dual-mode theory (Ekkekakis, 2003). Thank you for pointing out that this was not yet clear enough in the manuscript. We tried to improve this section by describing the theoretical claims concerning the ventilatory and respiratory compensation threshold in more detail and added a reference (review for empirical evidence).

Page 3:

“Dual-mode theory [9] explains how feelings during exercise are moderated by exercise intensity. According to the theory and supported by evidence [10], the affective response to moderate intensity exercise (below ventilatory threshold; VT) is mostly positive, but affective responses to heavy intensity exercise (approaching the VT) are more variable”

- Page 5, Line 9 – I am not sure I agree with the comment that biometric data is more unobtrusive. I think videoing someone’s face and facial analysis in general is just as obtrusive; it is just different than techniques researchers have traditionally used before. I think it is more powerful, but not more unobtrusive. Would like to see this statement rectified.

We agree that videoing someone’s face can also be obtrusive in a more general sense. Therefore, we now specified what exactly we mean by “unobtrusive” (i.e., by using this method ongoing physical exercise will not be interrupted e.g. by asking questions). We have added a respective passage in the limitations section, and revised all other passages in the manuscript were we had previously used the word “unobtrusive”.

e.g., Page 5:

“Monitoring changes in biometric data avoids these interruptions and can thereby provide an alternative way to learn about the feelings that occur during exercise.“

- Page 6, Line 7 – FEMG should be fEMG

We corrected this.

- Page 6, Line 11 – I don’t think you need to add an abbreviation for facial feature tracking as FFT. It wasn’t done in the original paper, isn’t commonly abbreviated and you don’t talk about it much after this.

We deleted the “FFT” throughout the manuscript and used the unabbreviated expression.

- Page 7, Line 18 – That first statement is quite strong, and I believe there are more than two processes that have been used. Maybe tone it down a bit and just say, “the most commonly used are…”

We toned the statement down by writing that “the majority of studies conducted so far has quantifies facial action by using…” 

- Throughout introduction (and the whole paper), you use facial action, facial expression and facial configurations interchangeably. I would just choose one, in your case it seems to be facial actions (especially considering you define why you are using this, which is good), and be consistent throughout.

Thank you for pointing that out. We now consistently refer only to "facial actions" and no longer to “facial expression”.

- Generally, I am not sure about the structure of the introduction. The reader must read through nearly eight pages of background information before you get to the project aim. I think that some of this background content is of minor consequence to the rest of the paper, quite repetitive and could be made significantly briefer. You want the reader to get straight into the sections that are really related to the content of your paper, like parts of 1.3 and all of 1.4. For example, the following sections don’t really relate strongly to the paper and could be removed or shortened significantly:

o Page 3, Section 1 – except for the definition of affect on lines 8-12

o Page 4, Section 1.1

o Page 6, Paragraph 4

o Page 7, Paragraph 1 and 2

Many thanks for this advice. We agree. We have restructured the line of argument and completely revised the introductory part of the proposed article. In particular, we deleted all unnecessary and redundant passages (and we devoted special attention to the text passages indicated by the reviewer above). We hope that the reviewer can understand how we proceed now: We try to efficiently guide the reader to the empirical part of our study, but to adequately tie in with the psychological literature from that research area (exercise psychology, affective responses to exercise) at the same time.

Method and materials

- Page 10, Line 1 – the use of the term modern is unnecessary. It is just software.

We removed the term “modern”.

- Page 10, Line 4 – what do you define as fit? Do you have any information on training history or sports played? Did you perform any baseline cycling assessments?

We no longer use the term "fit" and instead try to characterize the sample more accurately in other words. We added information about the self-reported weekly physical activity (e.g.; “The group average of (self-reported) at least moderate physical activity was 337 minutes per week.” ) 

Unfortunately, more than the information which is now reported in the manuscript is not available to us, and we have not made a cycling basement assessment. Anticipating the concerns of the reviewer, we have added restrictive comments in different parts of the text (in the methods section and in the discussion section).

For example page 26: 

“it would be advantageous to determine exercise intensity physiologically at the level of the individual participants in future studies (e.g., by the use of respiratory gas analysis in a pretest).”

- Page 10, Section 2.2 – Can you report participant height and weight?

Unfortunately, we did not measure height and weight of the participants. 

- Page 10, Line 7 – Why was the video quality poor?

The video quality was poor when the face detection algorithm could not identify a face, thereby leading to a high number of missing values. Participants with more than 10% missing values were excluded from the analysis. We included this more detailed information now in the methods section. 

- Page 10, Line 8 – What happened in the environment to cause this?

Other people, unrelated to the study, entered the room, and loud music from a gym next door has disturbed a handful of measurements. We included these details now in the method section. 

Page 9:

 (n = 6; more than 10% missing values because the software did not detect the face) or due to disturbing external circumstances (n = 7; people entering the room unexpectedly; loud music played in the gym).

- Page 10, Line 10 – “after one minute of baseline measurement” Baseline measurement of what?

Baseline measurement of heart rate. We have included this specification in the article now. 

Page 9:

“A Shimmer3 ECG device with a sampling rate of 512 Hz was used for that. These participants started with a one-minute heart rate baseline measurement before the exercise.”

- Page 10, Line 10-12 - A reference for the power output selection would be good.

We selected this power output according to the recommendation of Trappe & Löllgen (2000). This reference is included in the text now. 

- Page 10, Line 13-15 – “If a participant reached 300 watts the final phase involved pedaling at this level for two minutes, thus the maximum duration of the exercise was 26 minutes.” Why did you do this? How many participants completed the final stage without reaching volitional exhaustion?

Including an upper time limit was required by the ethical committee that finally approved this study, because the committee members thought that this would help to protect the physical safety of the participants. 11 participants reached the final stage of 300 watts, with n = 7 reporting an RPE of 20 and n = 4 an RPE of 19 at the end. According to their self-reports, all participants have reached volitional exhaustion; but of course we cannot be sure that all of them (as hoped for) have worked up to close to the state of actual physical exhaustion. We have revised the manuscript in several places in order to describe this more carefully now. For example (page 26):

“it would be advantageous to determine exercise intensity physiologically at the level of the individual participants in future studies (e.g., by the use of respiratory gas analysis in a pretest). This would have given us more confidence as to whether the majority of our participants have actually reached a state close to physical exhaustion at the end of the exercise protocol.”

- Page 10, Line 17-19 – Why did only half of participants have heart rate monitors?

You are right, that it could have been beneficial to record heart data for all participants. Unfortunately, we can not change this anymore. We decided beforehand to record heart rate only as a control variable, to show that the exercise protocol elicited an increase in heart rate and that the heart rate data corresponded with the RPE ratings (r = .82). We now try to make this clearer in the respective passage in the manuscript.

“For a plausibility check whether self-declared physical exhaustion would be at least close to the participants’ physiological state heart rate during exercise was monitored in about half of the participants (n = 54). (...) The correlation between heart rate and RPE was very high, r = .82, p < .001. “

 - Page 11, Section 2.3.3 – I think this would be a really good section for a diagram showing the points on the face or a technical schematic of how the software works.

We agree with the reviewer that a figure depicting Affectiva-landmarks on the face is helpful for a better understanding of how the software works. For this purpose, we replaced the pictures used in Figure 2 (now Figure 1), which now illustrates the algorithm’s identified and tracked landmarks as well as prototypical configurations of the facial actions “Nose Wrinkle”, “Jaw drop” and “Mouth Open” in an exercise-related context.

- Page 11, Section 2.3.3 – I think it would be good to have at least two cameras to pick-up different face angles. Is there a reason why you didn’t implement this? Do you think this affected the data you obtained during higher intensities when the head can drop? I think it is also important to know the distance the camera was to the face, an idea of head to camera angle, if it was adjusted for different participants height on the bike etc.

The face detection algorithm (Affectiva) as implemented in the iMotions biometric platform is technically restricted (but in the same way: optimized) to process signals from only one video source (camera). We agree with the reviewer that two cameras might lead to even better results. However, our study illustrates that using only one camera works sufficiently well: Only 6 participants had to be excluded from our study data set due to missing values (i.e. suboptimal face recognition and facial action analysis, according the algorithm’s error reports). 

We agree that some more information about the specific setup would be beneficial and added this specification to the methods section (p. 10):

“The camera was mounted 0.4 m in front of the face with an angle of 20 degrees from below, as it recommended by iMotions.” 

- Page 12, Section 2.3.3 – I think you need to describe more about the time each video frame was analysed for (i.e. milliseconds, seconds, batched into bigger 10 second blocks). At present it is not clear exactly how this was done.

We have revised section 2.3.3 and described more about the time each video was analyzed for.

“The algorithm performs analysis and classification at frame rate. This means that at a time resolution of 30 picture frames per video second (30 fps), our analyses were based on 1.800 data points per facial action per 1 minute.”

Frame-by-frame raw data was aggregated by using threshold analysis, which is described in more detail in response to the next question.

- Page 12, Section 2.3.3 – What is threshold analysis?

Thank you for pointing out that this point had not been sufficiently clear in the manuscript so far. iMotions identifies, tracks and analyzes facial landmarks continuously at picture/video frame-level (i.e., based on this frame by frame analysis iMotions provides facial action scores by threshold analysis). We tried to improve the respective sections (on the whole process and threshold analysis specifically) in the manuscript as follows: 

“Each data point that Affectiva Affdex provides for a facial action is the probability of presence (0 - 100%) of that facial action. We aggregated these raw data, for each facial action separately, to facial actions scores (time percent scores) indicating how long during a watt level on the ergometer (i.e., within 2 minutes) a facial action was detected with the value 10 or higher. For example, a facial action score of 0 indicates that the facial action was not present during the watt level, whereas a score of 100 indicates that it was present all the time during that watt level.”

- Page 12, Section 2.4 – So the participants knew the purpose of the video camera? I have a bit of an issue with this as I think this substantially changes how a person reacts to the video camera. Please address this as a limitation in your discussion. Might also be worth adding a reference for this manipulation of facial action when the participant knows their expression is being monitored. For example, this reference: Philippen P, Bakker F, Oudejans R, Canal-Bruland R. The Effects of Smiling and Frowning on Perceived Affect and Exertion While Physically Active. J Sport Behav. 2012;35(3):337-352.

According to ethical reasons, the participants were informed that their faces will be filmed during the study. After the experiment, we asked them whether the camera had affected their behavior. Participants responded to this question on a scale from 1 (“do not agree at all”) to 5 (“fully agree”) with a mean of 1.57 (SD = 0.69), indicating that the camera did not influence their behavior substantially. We did not include this information in the manuscript because the participants could still have been biased by the presence of a camera. Therefore, of course, we agree with the reviewer’s comment that video observation can sometimes influence behavior without the participants noticing it themselves.

In order to address the reviewer’s comment we have added the following sentence with the mentioned reference in the limitations section of the manuscript and referenced the suggested study (thank you for bringing it to our attention):

Page 25:

“It is important not to lose sight of the fact, however, that simply knowing that you are being filmed can of course also change your behavior [52]. “ 

Results

- Page 14, Line 13 – You are introducing some commentary and interpretation into your results, which I would suggest removing or putting in your discussion

We agree with the reviewer that the interpretation of study results must be part of the discussion. We decided to delete this interpretation (that differences in the reached intensity stage were probably due to differences in physical fitness) from the results section.

- Page 14, Lines 17-19 - Based upon the fact that HR was only recorded for half participants and they only reached a mean HR of 174.61 and that RPE was 19 with an SD over 1 which suggests a large number of participants reported RPE well below maximum, I really don’t think you can say that for this age group they reached maximum capacity. Please alter statement.

It is indeed possible that not all of the participants have reached maximum capacity, and we have therefore included a note on this in the limitations section (p. 26). However, with respect to previous findings, we still believe that our data and findings are plausible: For example, Roecker, Striegel and Dickhuth (2003) found an average HRmax of 179.5 ± 20.2 for 129 recreational sports participants, which is close to our reported average HRmax of 174.6 ± 16.1 (we incorporated this reference in the manuscript). Importantly, subjects would usually reach lower maximal heart rates on cycling ergometers compared to when they are asked to run on a treadmill. We have included this reference in the manuscript.

- The image quality and clarity of Figure 1 is poor and does not convey the point that I can see you want to get across very well. Maybe think of an alternative way to represent this data?

We agree that the image quality and clarity of Figure 1 (now Figure 2) is poor in the submission pdf. PLOS One informed us that the compiled submission PDF only includes low-resolution preview images, to allow you to download the entire submission as quickly as possible. By clicking on the link at the top of each preview page, it is possible to download the high-resolution version of each figure. We submitted our figures according to journal requirements and double-checked with the PACE-tool (as suggested on the journal homepage). 

We prefer to keep Figure 1 (now Figure 2) in the manuscript. Its purpose is to illustrate the considerable intra- and inter-individual variability in the participants’ affective responses to increasing exercise intensity. Including this figure underpins the (in our view) necessity of using multilevel regression models for analyzing data of this kind (previous studies had used standard regression analysis that ignores individual differences by aggregating individual slopes). In order to strengthen this point, we included the following passage in the results section with regard to Figure 2 (p. 14):

Fig 2 illustrates the finding, which can be made particularly obvious by means of multilevel regression analysis: The high interindividual variability in the decrease of affective valence (more negative) under increasing perceived exhaustion is striking.

Discussion and conclusion

- Page 23, Line 11 – you refer to the “aerobic exercise” here. This I the first time you refer to it as this, which is a bit odd. Maybe better to keep it consistent to what you said in the introduction, such as an incremental test.

We corrected this term according to your recommendation as “incremental exercise test”. 

- Page 23, Line 13 –you use e.g. mid-sentence again. Will flow better if written as for example or something similar in the sentence structure.

We corrected this and used “for example” instead.

- Page 26, Section 4.6 – Generally, you don’t report any of the limitations that this study appears to have in your limitations section. Some of the main ones I have noted above are the fact participants knew the purpose of the video cameras so could change expressions, you didn’t obtain HR for all participants so cannot determine if they did actually reach the intensities you aimed for, and the final exercise intensity was absolute and fixed to 300W. Please rectify.

Thank you for pointing that out. We have thoroughly revised the limitations section and included the points mentioned by the reviewer (p. 26; the influence of knowing to be filmed):

 “It is important not to lose sight of the fact, however, that simply knowing that you are being filmed can of course change your behavior [52].

Limitation in statement about exercise intensity:

“It would be advantageous to determine exercise intensity physiologically at the level of the individual participants in future studies (e.g., by the use of respiratory gas analysis in a pretest). This would have given us more confidence as to whether the majority of our participants have actually reached a state close to physical exhaustion at the end of the exercise protocol.”

- Page 26, Lines 7-9 – This is a strong statement. I would just tone it down a bit…

We removed this strong statement.

- Page 26, Lines 10-11 – I like this section and agree with what you are saying, but I don’t feel the methods section you have provided in this paper would allow this research to be expanded upon by others. I think an improved methods section could be the strength of this paper

Thank you for expressing your appreciation. We tried to improve the methods section according to your comments and hope that it now allows other researchers better to replicate our work and findings (or build up on them). 

- General comment throughout discussion and conclusion– You discuss that mouth open and draw drop are the “face of exertion”. I don’t disagree with this statement per se, and yes, it is shown in your results. However, I feel you need to more strongly note that as RPE increases, someone is likely to be breathing heavier, and therefore the jaw drops and mouth opens. At present you don’t really discuss this in any detail, which I feel is an oversight in the interpretation of your findings.

Thank you for this valuable addition to a more adequate interpretation of our findings. We emphasized this issue in greater detail now on p. 24: 

“Both the metabolic thresholds, VT and RCP, are related to perceived exertion. They are objective, individualized metabolic indicators of intensity, and are already associated with psychological transitions in dual mode theory [9]. Linking them to transitions in facial actions could be a future prospect and be something like this: While exercising at the VT might mark the transition between nose to (predominantly) mouth breathing and thus also the transition to more mouth open, exercising above the VT might mark a transition to more jaw drop. This kind of intensified breathing might covary with escalating negative affective valence – that is the evolutionary built-in warning signal that homeostatic perturbation is precarious behavioral adaptation (reduction of physical strain) is necessary [12].“

Reviewer 2

Reviewer #2: Review of manuscript PONE-D-19_26758 submitted to PLOS ONE

In this study, facial expressions during aerobic exercise were recorded at fixed time intervals using an algorithm to detect specific facial actions and identify them as actions units defined within the Facial Action Coding System (FACS). These expressions were related to subjective ratings of positive or negative affective state as well as ratings of perceived exertion. Covariations between facial expressions and subjective ratings were analyzed using multilevel regression or trend analysis, allowing investigation of these relationships at the level of individual participants.

The rationale behind this study is clear, the statistical analyses are sophisticated, and the manuscript is well written. I agree with the authors that a better insight in the exerciser's affective state and subjective feelings of exertion may contribute to stimulating people to engage in further pleasurable physical exercise. Although the results of this study seem clear, I have a few comments, questions, or (generally minor) concerns.

Thank you for sharing your thoughts with us! We have tried to take up all points raised by the reviewer and try to improve the manuscript. Before we start discussing the single points, please let us politely correct one aspect of the summary above, because we believe that some of the reviewer's comments below will be easier to answer after having done so:

Facial actions during exercise were recorded continuously and Affectiva’s Affdex algorithm was used to analyze facial actions. This algorithm statistically compares observed changes with data from a normative database with information from more than 6 million faces of participants from 75 countries. The software records and continuously analyzes the configuration of 34 facial landmarks, with a resolution of 30 frames per second (fps). The Affdex algorithm returns as a sequential score at frame rate in real time (and stores for later analysis) probabilistic results (0–100%) that indicate the likelihood of occurrence of defined facial actions (Affectiva, 2018). Importantly, the Affdex algorithm involves elements of artificial intelligence (nonparametric machine learning), so that more detailed information about the transformation of facial action data into scores for facial expression is not available. This is a major difference to the use of the FACS, which is based on parametrically defined coding rules. 

In fact, the Affectiva algorithm was initially trained with data based on human FACS-coding. As the result of the algorithm’s training however, Affectiva deviates from FACS in a few aspects (e.g. the Affectiva “smile” score is formed from a combination of FACS AU 6 + AU 12). A detailed descriptions of the facial actions provided by Affectiva can be found on https://developer.affectiva.com/metrics/. We included this reference now also in the relevant passages throughout the manuscript.

1. On pp. 6-7, and in the remainder of this manuscript, emphasis is laid on facial actions as indices of affective states or specific emotions. However, facial actions may also be related to perceptual, motivational, attentional, or cognitive processes (see, for example, studies mentioned in reference 58 of the current manuscript; see also Overbeek et al., 2014, Stekelenburg & van Boxtel, 2001). Although in the current study a lot of different facial actions were measured it seems as if the authors a priori consider these actions as indices of emotional processes whereas within FACS individual actions units strictly do not refer to specific emotions.

Thank you for this valuable remark. We agree that facial actions can be related to different processes and not just specific emotions. Therefore, we have incorporated the notion that facial action may also be related to perceptual, motivational, attentional, or cognitive processes in the manuscript with your suggested references.

For example, see p. 5:

“However, it cannot universally be assumed that observed facial movements always reflect (i.e., are expressive of) an inner state [16]. Facial actions can also be related to perceptual, social, attentional, or cognitive processes [17, 18].” 

2. Although the current software used for analyzing facial actions indeed detected elementary facial actions as indicated in Table 1, it is remarkable that one of these actions ("smile") does not represent an elementary action but a combination of actions (AU6, check raiser; AU12, lip corner puller). This combination is generally interpreted as signifying a smile. I find this confusing since emotions are strictly not measured in the current study. On p. 25, it is said that in this study smile was associated with a negative affective valence. In line 12 on this page, it is erroneously suggested that the detection of AU6 is synonymous with the occurrence of a smile.

We agree with your comment that different labels in FACS and Affectiva scores are confusing and our presentation of this was confusing in the first version of the manuscript. We have now tried to improve this (make the difference between FACS and Affectiva even clearer in the manuscript) and to avoid (even more; further) confusion.

In conclusion, we think that we have to stick to the nomenclature and classification system in Affectiva (https://developer.affectiva.com/metrics/) because only this terminological system was used in this study. That is why we continue to use the labels (e.g. “smile”) provided by iMotions Affectiva so that further researchers working with the same system (Affectiva) can relate their findings to ours.

Last but not least, thank you for your drawing to our attention, that we erroneously wrote that smile is solely the occurrence of AU 6. We corrected this on p. 25. 

3. Figure 2 illustrates relevant facial actions which were observed during the current physical exertion task. I find these examples somewhat confusing since for the reader it may be difficult to associate them with an aerobic exercise task. But particularly the illustration of jaw drop is confusing since this facial expression also depicts AU's 6 and 12 which are generally considered to represent happiness, suggesting that this person is overtly laughing. I have shown this picture to several colleagues asking them to indicate what they saw. They reported to see an overtly laughing person.

Thank you for bringing this to our attention! We agree that these pictures can be confusing. We have deleted them and included context-specific material now (see Figure 1), i.e. example pictures during exercise, which should allow a much better illustration of the points that are important to us (showing high Affectiva-scores for “Nose Wrinkle” and “Jaw Drop” in an exercise context).

4. On p. 23, it is defended that nose wrinkle need not be specifically related to disgust and that it may also be indicative of other emotions. However, in this respect studies are mentioned which have been performed in infants. I am afraid that facial expressions of infants cannot directly be compared with those of adults.

Thank you for mentioning this important point. We can well understand your concerns that studies with infants should not be generalized to our participants. Instead, in the manuscript we now focus previous studies with adults, and refer to the there-discussed association between “nose wrinkle” and “pain” ( references 54, 55), and we have edited the respective passage as follows:

“Nose wrinkle has also been specifically associated with the emotion disgust [15]. However, the same facial action has been observed in various situations (e.g. while learning) [46] and different emotional states (e.g., anger) [48].”

5. Later on this page, it is concluded that mouth open and jaw drop are highly correlated with perceived exertion but that this does not agree with results from an EMG study which would suggest that perceived exertion during physical tasks is mainly linked with corrugator activity. This brings me to the general question whether discrepancies between different studies may (at least partially) be related to studying either aerobic or anaerobic exercise. This distinction is not really discussed in this manuscript. When suggesting on p. 26, third paragraph, that future studies should include a wider ranger of sports to assure a higher generalizability of the current results, I wonder whether types of anaerobic exercise shouldn't also be included.

Thank you for that point. We focused mainly on cardiovascular load in an incremental endurance test to study the affective response against the background of dual mode theory, and tried to explain that better in the manuscript now. We agree with the reviewer that different forms of exercise could be associated with different facial action responses and that further research should address this. We have now written in the manuscript (p. 26):

“Future studies should extend the use of automated facial action analysis to a wider range of participants and sports to assure higher generalizability of the findings reported here. Different modalities and different exercise intensities might produce specific facial actions. More heterogeneous samples are likely to produce more variance in affective responses, which may lead to further insight into the variation in facial reactions to exercise. 

The current study is limited in drawing conclusions regarding differences in the affective response to different intensity domains as exercise intensity was not measured physiologically.”

Minor points

- P. 7, line 3: "action" > "actions"

- P. 9, line 10: "action" > "actions"

- P. 11, line 11: "Logitech HD Pro C920" > "Logitech HD Pro C920 webcam"

- P. 19, footnote to Table 2: "in less number of parameters" > "in a smaller number of parameters"

Thank you for pointing this out, we corrected all these points in the manuscript.

---

## [Editor Report · Decision Letter 1]

23 Jan 2020

Affect and exertion during incremental physical exercise: Examining changes using automated facial action analysis and experiential self-report.

PONE-D-19-26758R1

Dear Dr. Timme,

We are pleased to inform you that your manuscript has been judged scientifically suitable for publication and will be formally accepted for publication once it complies with all outstanding technical requirements.

With kind regards,

Dominic Micklewright, PhD CPsychol PFHEA FBASES FACSM

Academic Editor

PLOS ONE

---

## [Editor Report · Acceptance letter]

28 Jan 2020

PONE-D-19-26758R1 

Affect and exertion during incremental physical exercise: Examining changes using automated facial action analysis and experiential self-report. 

Dear Dr. Timme:

I am pleased to inform you that your manuscript has been deemed suitable for publication in PLOS ONE. Congratulations! Your manuscript is now with our production department. 

With kind regards,

on behalf of

Professor Dominic Micklewright 

Academic Editor

PLOS ONE